# Parents’ and Health Professionals’ Attitudes to Advancing Primary MMR Vaccine Administration from Fifteen to Six Months of Age—A Qualitative Thematic Analysis Embedded in a Randomized Trial

**DOI:** 10.3390/vaccines11010067

**Published:** 2022-12-28

**Authors:** Ann-Britt Kiholm Kirkedal, Julie Elkjær Møller, Lone Graff Stensballe, Vibeke Zoffmann

**Affiliations:** 1Department of Pediatrics and Adolescent Medicine, Copenhagen University Hospital, Herlev and Gentofte Hospital, 2730 Herlev, Denmark; 2Department of Pediatrics and Adolescent Medicine, Copenhagen University Hospital, Rigshospitalet, 2100 Copenhagen, Denmark; 3The Interdisciplinary Research Unit of Women’s, Children’s and Families’ Health, The Julie Marie Center, Copenhagen University Hospital, Rigshospitalet, 2100 Copenhagen, Denmark

**Keywords:** MMR vaccine, parental attitude, decision making, self-determined choices, vaccine supporters, vaccine opponents, trust, healthcare professionals

## Abstract

Declining levels and duration of passively acquired maternal antibodies prompted a Danish trial to test the feasibility of advancing administration of the first measles, mumps, and rubella vaccine (MMR1) from 15 to 6 months of age. A trial-embedded qualitative study aimed to understand parents’ (*N* = 24) and health professionals’ (*N* = 11) attitudes about the measles, mumps, and rubella vaccine (MMR) in general and about advancing MMR1 administration. Overly positive parent attitudes were contrasted by members of a vaccine-skeptical organization including parents considering that their child was seriously vaccine-injured long ago. Parents’ attitudes to advancing MMR1 mirrored their attitudes about the MMR vaccine in general, with four positions along a continuum of trust in the healthcare system: unquestioning trust, acceptance after careful consideration, challenging indecisiveness, and defensive rejection. Low tolerance was identified between vaccine supporters and vaccine opponents. Parents of children with perceived serious vaccine-related injuries described lifelong unresolved feelings of guilt. Supporters of advanced MMR1 saw it as a timely and convenient administration of a well-known vaccine, whereas opponents feared it would disturb the children’s immature immune systems and emphasized difficulties in recognizing side effects so early in life. Health professionals were supportive of advancing the MMR1 vaccine and they carefully challenged the parents. Current MMR vaccine supporters show readiness to advance MMR1 administration.

## 1. Introduction

In Denmark, the combined measles, mumps, and rubella vaccine (MMR) was introduced in 1987 as part of the childhood vaccination program offered at no cost to all children in Denmark [1]. MMR administration is scheduled at fifteen months (MMR1) and 4 years of age (MMR2). However, due to declining levels and duration of passively acquired maternal antibodies [2], a Danish trial was conducted to test the feasibility of advancing MMR1 administration from 15 to 6 months of age. Beginning in April 2019, a double-blind, randomized, and placebo-controlled clinical trial (“The measles-mumps-rubella vaccine at 6 months of age, immunology, and childhood morbidity in a high-income setting trial” (MMR trial; EudraCT 2016-001901-18; Clinicaltrials.gov NCT03780179)) enrolled 6500 infants to examine whether MMR1 could be administered at 6 months of age [3]. The expected study findings include specific immunogenicity of an advanced MMR vaccine, adverse events, and potential beneficial, non-specific effects of the vaccine, which will be published in 2022–2024.

Although they are pivotal to developing a successful change strategy, parents’ and health professionals’ attitudes toward advancing MMR1 administration have rarely been investigated. Countries in the World Health Organization (WHO) European region have committed to eliminating measles and rubella [4]. High vaccination coverage rates are crucial in stopping the spread of measles. Before 1987, 98% of the population had contracted measles in Denmark, a proportion that dropped dramatically after the introduction of the MMR vaccine. In 2019, the uptake of MMR1 in Denmark was 94% [5]. Despite the relatively high adherence to the MMR vaccine, measles outbreaks have been observed with increasing frequency in high-income countries, including Denmark [6,7]. To increase vaccine coverage, the Danish government allocated funds for a four-year plan involving new immunization initiatives, beginning in 2019. Initiatives include training health visitor nurses (HVN) as immunization ambassadors and a new online reminder system [7].

Acceptance of the childhood vaccination program in Denmark is high; 96% of children receive the combined diphtheria, tetanus, pertussis, polio and hemophilus vaccine and the pneumococcal vaccine. However, population coverage for MMR1 (94%) and MMR2 (89%) fail to meet the WHO-recommended 95% [5,7]. In 2019, the WHO specified vaccine hesitation as among the top ten threats to global health [8]. After its introduction, the MMR vaccine engendered many discussions and more concerns than other childhood vaccinations [9], until the suggested risk of autism was clearly refuted [10]. 

Internationally, qualitative studies have investigated reasons why parents choose to postpone or reject the MMR vaccine, which are related to concerns about its safety, including fear of side effects [11] or perceived relative harms of the vaccine and the diseases it prevents [9,12,13,14]. Other concerns relate to how best to stimulate the immune system in children [11,13,15,16]. Mistrust of healthcare authorities also influences MMR vaccine acceptance [17,18]. In 2006, a Danish qualitative study based on semi-structured interviews with 17 parents elucidated similar reasons why children in Denmark did not receive the MMR vaccine. Parents expressed concerns about side effects, a desire for natural disease-acquired immunity, risks related to combined vaccines, and fear that the vaccine would harm the child more than the diseases would. Mistrust was also expressed as a perception that health authorities vaccinated children to avoid parental absences from work to care for ill children [19]. During the COVID-19 pandemic, the debate about vaccine acceptance or hesitancy increased [20]. 

Although parental concerns about MMR vaccination have been partly elucidated by the previous studies, the ongoing trial of earlier MMR1 administration provides a rationale for a new investigation to support successful implementation of positive trial results. The study purpose was to investigate parental and healthcare professional (HCP) attitudes about MMR vaccination in general and to earlier administration of the primary MMR vaccine. 

## 2. Materials and Methods

A qualitative study embedded in the MMR trial was initiated after participants had provided final quantitative data. The study was designed as a qualitative interview study followed by an inductive thematic analysis based on the recommendations by Braun and Clarke [21,22].

### 2.1. Participants

#### 2.1.1. Parents

Twenty-six parents were purposefully sampled [21], comprising 19 women and 7 men. The first six parent participants (parents 1–6, Table 1) were recruited at the MMR trial site, had children aged five to six months, and were expected to view MMR vaccination positively. To obtain more diverse perspectives, we also recruited parents who were not invited or had declined to participate in the MMR trial. Moreover, we contacted two websites: younger doctors inform the public about different health issues on “Læger Formidler” and a private HVN who answers a wide variety of parental questions about infants and children, including inquiries about vaccinations on “Netsundhedsplejersken”. Our contact information was available on the websites, and parents were invited to contact us by phone or mail (parents 7–19, Table 1). All parents who contacted us and wanted to participate through these sites were pro-vaccination or undecided. Thus, we specifically searched for parents who were opposed to vaccination and, with permission, approached a family member of one of our colleagues who had agreed to participate (parent No. 20, Table 1). 

To recruit more opponents of MMR vaccination, we placed posters in the HVN office and at the local pediatric department. We also approached participants 12 and 21 (Table 1), who were undecided and anti-vaccination, respectively, and asked if they could help us reach more parents who shared their points of view. These efforts were unsuccessful. However, due to public awareness of the MMR trial, a group of parents (parents 21–24, Table 1) who knew each other through an anti-vaccine organization contacted us. Despite the fact that their children were much older than those of our primary group of parents, we included them to understand resistance against vaccines, the atmosphere between vaccine supporters and opponents, and the long-term perspectives of two parents whose children were believed seriously injured in connection with MMR vaccination 27 and 17 years before.

#### 2.1.2. Healthcare Providers 

Danish healthcare, including childhood vaccinations, is free of charge; vaccinations are recommended by the National Board of Health (NBH) but not mandatory. HVNs visit families with newborn babies several times during the first year. They examine the child’s physical, mental, and social health and talk to parents about healthcare, nutrition, and the family [23]. All families are offered examinations of their children at five weeks, five months, and annually at ages 1–5 years by their general practitioners (GPs), who administer all childhood vaccinations. Six HVNs were recruited by contacting HVN centers in Copenhagen. Four GPs were recruited, three through a representative for GPs who is elected to bridge collaboration between GPs and hospital wards. A fourth GP and a nurse working at a GP’s office were recruited due to prior collaboration with V.Z.

### 2.2. Data Collection

All interviews with parents were conducted as in-depth semi-structured interviews [24] in June 2020–January 2022. All but two interviews, in which both parents participated, were individual. Baseline characteristics, including age, marital status, educational level, and number of children in the family were recorded at the time of interviews. Interviews with healthcare professionals (Table 2) were conducted in a focus group, a dyad and individually at the participants’ request. All interviews were planned to take place in person, but the social restrictions introduced to limit COVID-19 transmission prohibited face-to-face interviews and necessitated conducting many by videoconference or telephone. Interviews with parents participating in the MMR trial were occasionally slightly abbreviated because their children had just received a medical examination, a vaccination and were fatigued.

Interviews were semi-structured, with iterative changes to the interview guide over the course of data collection. For instance, we further explored the topic of trust in the healthcare system because it frequently arose during interviews. 

Parents were primarily asked about their attitude toward the MMR vaccine and advancing MMR1 administration from fifteen to 6 months. We also posed questions about concerns and challenges that arose during decision making about the MMR vaccine in general and advancing MMR1 administration. Moreover, we asked what kind of information they believed would be necessary to accept an earlier MMR1 vaccination and where they currently sought that type of information. All interviews with parents were conducted by A.-B.K.K. or J.E.M. and lasted 16–38 min with most parents. Interviews with the parents of children with MMR-related injuries lasted 66–83 min. All interviews were recorded and transcribed verbatim by A.-B.K.K., J.E.M., and a medical student. 

All HCPs were asked how they usually handled parental doubt about or rejection of the childhood vaccination program. They were asked questions about parents’ attitudes towards the MMR vaccination and how they usually addressed parental concerns. In addition, HVNs were asked questions about their new role as immunization ambassadors and how this affected their daily work and relationships with parents. HCP interviews lasted 42–46 min.

The study was registered in the capital region’s data protection authority (H-16041195), and all participants signed an informed consent form after receiving verbal and written information about the purpose of the interview. Participants received no financial compensation.

### 2.3. Data Analysis 

Interview data were analyzed systematically following the six steps recommended by Braun and Clarke [21,22] for inductive thematic analysis: (1) familiarization with the data, (2) generating initial codes, (3) searching for themes and gathering data relevant to each potential theme, (4) reviewing themes and generating a thematic map of the analysis, (5) defining and naming each theme, and (6) producing the report. Consolidated criteria for reporting qualitative research were followed [25]. 

A.-B.K.K. and J.E.M. transcribed the interviews and familiarized themselves with the entire data set, reading it several times to obtain an overall understanding of the material. In close dialogue with V.Z., they then generated initial codes and conducted continuous analysis of the data inductively and iteratively going back and forth between meaning units, subthemes, and preliminary themes until substantial themes were identified. Finally, A-B.K.K., J.E.M. and V.Z. discussed the findings with L.G.S. until consensus was reached, strengthening the validity of the analysis. We analyzed data from parent interviews first, followed by data from HCP interviews. Some themes arose in both groups, enriching our understanding of the themes.

## 3. Results

We identified three themes in data from parent interviews. A primary theme differentiated four attitudes to the MMR vaccine in general and to advancing MMR1 administration that were located along a continuum of parental trust in the healthcare system (Table 3): (1) unquestioning trust, (2) acceptance after careful consideration, (3) challenging indecisiveness, and (4) defensive rejection. Secondly, a theme of low tolerance between vaccine supporters and vaccine opponents was identified. Thirdly, lifelong unresolved feelings of guilt were identified among parents of seriously injured children. In data from HCP interviews, only positive attitudes were found about the MMR and advancing MMR1 administration. A fourth theme was identified, characterizing HCPs’ response to vaccine-reluctant parents: challenging vaccination reluctance but careful to avoid losing contact with families.

### 3.1. Parental Attitudes about MMR and Advancing MMR1 Administration

#### 3.1.1. Unquestioning Trust

We expected a positive attitude among parents participating in the MMR trial, but the degree to which the attitudes of some parents were unconditionally positive was surprising. Some even had difficulty explaining their position. One mother explained her obvious positive view on having her child vaccinated as follows:


*That’s just like such a thing that she [the child] has to wear shoes and diapers, she also has to go to the doctor and be weighed and measured and vaccinated.*
(Parent 19)

Family traditions were often mentioned, and the decision was not questioned:


*That’s how I was raised. I’ve also gotten it myself. So that is what you do [get the child vaccinated].*
(Parent 4)


*If anyone says this [recommending vaccines], then it must be because it is true.*
(Parent 13)


*We haven’t even discussed that she shouldn´t have it.*
(Parent 5)

In general, parents with unquestioning and unconditional vaccine acceptance had a completely trusting attitude about the healthcare system. They were not very active in seeking information about vaccines because they trusted the healthcare authorities and were convinced that the healthcare system related to vaccines worked well:


*I trust enormously the professionalism that is in our, what is it, our health board, which has worked on this for many years.*
(Parent 19)


*I think when the health authorities also recommend that this, it´s what you do. Then, I think that, then the risk of getting it is probably not as great in relation to what, what else you get out of it [the vaccination].*
(Parent 17)


*If there is evidence for it, then that is what we follow. Then, there is no reason to ask too many other critical questions. The professionals who know something guide us in the right direction.*
(Parent 2)

This position seemed uncomplicated for parents. Their attitude was seldom questioned by other people because most of their acquaintances had the same view and it was consistent with recommendations from their GP and HVN. 

#### 3.1.2. Acceptance after Careful Consideration

Another positive attitude was seen among parents who expressed their reasoning for and against the MMR vaccine. They sought out vaccine information, particularly from the NBH home page and provided a rationale for their decision that reflected a balanced and positive result of weighing the risks of disease versus those of the vaccines. This was expressed as below:


*I just like to go into it [NBH home page]. I thought it was very clear, and you can look up the vaccines separately.*
(Parent 9)


*I think that the side effects from a vaccine are, uh, relatively much less than having to go through these disease courses.*
(Parent 18)


*So, that makes it an easy decision for us. We will not expose our children to this [the MMR diseases] if we can avoid it.*
(Parent 14)

These parents were also positive about earlier MMR1 administration, emphasizing that it was a known vaccine and more convenient to get the child vaccinated at that time rather than later, in case of side effects. 


*For us, it’s a lot about the practical because uh, if they [the children] get side effects when would it fit in. We try like to put it like that uh the MMR vaccine such a week before we even went on holiday (…) if it so was at 6 months, so just see we need something practical there, but not so much beyond that.*
(Parent 14)


*Let’s now say it had been a new vaccine with MMR, so not just the old vaccine, but a new vaccine they tested, then I would have been more in doubt.*
(Parent 13)


*But I think it’s mostly because people are not properly informed. I also think that people have come so far away from the disease flourishing in society that now they do not think about it. It is almost a given that it is not a risk anymore.*
(Parent 8)

#### 3.1.3. Challenging Indecisiveness

Some parents were indecisive about whether their child should be vaccinated and found their position challenging: 


*But it’s just a little annoying to start out with such a skepticism. I would much rather just be able to blindly trust it [the NBH].*
(Parent 12)

This position was very difficult to maintain, particularly when it reached a stage of deadlock as expressed by the following mother. She let her child be vaccinated, a choice that she did not feel she had actually made:


*I was just very much in doubt and in the end, I actually felt that I really only chose to vaccinate him because I actually didn’t know what else to do. It was a choice in powerlessness. It sounds a bit wrong, but then it was a ‘non-choice’.*
(Parent 12)

This position was related to limited trust in the healthcare system. Although parents had often sought information, they had limited trust in its source.


*Then I cannot trust that the National Board of Health gives me objective information because they have an interest together with the Serum Institute to sell vaccines.*
(Parent 12) 

A challenge connected with this position was the criticism parents received from both vaccine supporters and vaccine opponents when they voiced their doubts. Consequently, a sense of isolation and the perception of not belonging anywhere were inherent in indecisiveness, as expressed by the same woman as in the previous quote:


*So, I actually think I’m a bit in between […] yes to both camps. Then there’s not really room for me anywhere in either.*
(Parent 12)

Parents in this position were concerned about earlier MMR1 administration because they feared it might disturb an infant’s immature immune system. 


*I wish his [her child’s] immune system was allowed to develop without being disturbed. If I had to vaccinate them [the children] at all, I would at least like them to be more than 3 years old, so their system was a little more well developed so we do not disturb it too much.*
(Parent 20)

Another argument against earlier MMR1 administration was that it might be risky when given so close to other childhood vaccinations:


*…that it [the MMR] should be given earlier, in the middle of the childhood vaccination program with the three vaccines there, I think it’s just completely in the wrong direction. If it is at 6 months you want to give it, then it is also only 1 month after the second vaccine. Yeah, so it just seems very careless, I think.*
(Parent 21)

#### 3.1.4. Defensive Rejection

Parents who rejected the MMR vaccine clearly expressed mistrust in the healthcare system and resistance to being told what to do. They seemed to feel overruled and confronted, as in the following quote:


*Doctors have been given kind of God status in Denmark… right? When the doctor says: “it’s good”, then it’s good, you know… I just wonder: why are you not able to be a little more critical?*
(Parent 20)

These parents voiced a strong desire to defend themselves and what they considered their free choice against the authorities:


*I just think in reality… it must be the right to a fucking free choice and respect around that choice. From the entire system.*
(Parent 23)

Parents who took this position also expressed doubt about the severity of the MMR diseases: 


*And I do not see that the diseases you are vaccinated against in Denmark are so horrible that you can’t just deal with it, again with a few exceptions.*
(Parent 23)

In keeping with this point of view, the woman below expressed support of the principles used before the MMR vaccine was available: namely, that having the MMR diseases during childhood was preferable. 


*I really wish they [the children] had some of the childhood diseases because I think it helps to mature and develop them, and there is a sense that we should have the childhood diseases while we are children well and truly.*
(Parent 20)

Some parents expressed disbelief in the information from the NBH and the Serum Institute. Accordingly, they pursued information about vaccines from alternative sources. Several emphasized that the literature was “in English” and ”written by doctors”. When we asked about specific literature or authors they relied on, only a few supplied names. Others talked in more general terms: 


*I have seen hours of lectures. I have more than 12–15 books [about the subject] written by scientists and doctors.*
(Parent 22)

This group of parents was highly negative about advancing MMR1 administration, due to what they perceived as the child’s immature immune system:


*Well, that [advancing MMR-1] sounds so awful. I really thought we should not disturb the little people too soon in their little bodies. It makes me really sad, that that is what you are aiming at.*
(Parent 20)

As reflected in her emotional response, this woman was seriously concerned about the influence on infants of an earlier MMR1. Other parents invoked the idea of ‘survival of the fittest’ as an unavoidable condition that the vaccination program mistakenly tried to overcome:


*Well, we [the healthcare system] are trying to make the weak stronger by weakening the strong. It is wrong. You don’t have to. The strong who are made stronger, they want to lift the weak and they should probably do it, I’m 100 on that. The good genes go on and the bad ones disappear. That’s how it is. Whether you like it or not.*
(Parent 23)

Another parent, a mother of two children, was convinced that her unvaccinated children suffered less from diseases than those who were vaccinated: 


*And I also think it’s striking that when I compare my unvaccinated kids with children vaccinated, oh my God! The amount of sick days—it’s striking. It is! It’s crazy the difference. The kids are all ailing.*
(Parent 23)

This point of view was supported by a parent who shared similar reflections:


*So I just found it interesting that many of those I know, who do not vaccinate their children, none of our children have anything at all with the immune system or allergies or eczema or anything. And pretty much everyone else I know [vaccinated children] gets sick.*
(Parent 12)

### 3.2. Low Mutual Tolerance between Vaccine Supporters and Opponent

Among supporters and opponents of MMR vaccine, feelings of frustration were directed to people who had the opposite point of view. 

Some parents supporting the MMR vaccine thus expressed strong feelings toward parents who were unwilling to have their child vaccinated. 


*…if I think too much about it, I become sort of really angry with those, who don’t do it [vaccinate], if they can.*
(Parent 5)

These parents even saw the need to legislate on the topic.


*If you choose not to vaccinate, then you can’t go to daycare and school—then you can’t get any public subsidy.*
(Parent 7)

Conversely, parents who opposed vaccines explained how their frustrated skepticism had been amplified by pressure and even ridicule by citizens and HCPs, a tendency they believed had been exacerbated over time.


*If you don’t vaccinate, you are just an “idiot”, right? [……] instead of just maybe ask about my thoughts about it.*
(Parent 20)


*Just the fact that you ask questions makes you unscientific, a “tinfoil hat”, you know, almost illiterate and worse.*
(Parent 23)

Feeling that authorities dictated whether one vaccinated one’s children thus appeared to influence these parents’ attitudes to people with the opposite point of view. For example, the sarcastic quote below demonstrates the frustration of a mother who complained that it was not OK for people to raise questions to health authorities.


*So, it is not okay to question what order the doctor puts out or what order the National Board of Health puts out, it is just not!*
(Parent 20)

### 3.3. Lifelong Unresolved Feelings of Guilt 

Lifelong unresolved guilt was described by parents of children perceived as seriously injured in connection with a past MMR vaccination. One parent critical of the MMR vaccine had used the phrase the “almost fully safe vaccine”. She was fully aware of the minimal risk. Yet, she pointed out that the statistically minimal risk “becomes 100% when you get hit” (parent 21).

The truth of this statement was evident in interviews with two parents who belonged to a vaccine-skeptical organization (parents 21 and 24). Their children had developed brain damage shortly after MMR vaccinations 27 and 17 years before, and are still struggling with severe developmental and behavioral disorders. Both sets of parents are convinced that the brain damage is due to the vaccine, and medical professionals have in the past made a thorough examination of the children’s medical records and found no other obvious reason for their difficulties. 

The families contacted both GPs and emergency rooms when their child became ill with high fevers and neurological symptoms within a few days after the first MMR vaccine. However, both parents described being met with distrust by HCPs when voicing their suspicion that the symptoms were caused by the vaccine. This gave them a feeling of being let down by HCPs:


*We said it must have been the vaccination, but the doctor said no, no, no, no. That can’t be right. I wouldn’t say that I that I feel looked down upon. However, I have felt that they didn’t believe me.*
(Parent 24)

They both described reluctance, even denial, from Danish doctors towards reporting vaccine side effects. It took both families many years to have their child’s difficulties recognized as resulting from vaccine. Significant feelings of guilt developed and were still present decades after the vaccination. Both mothers revealed how lifelong feelings of guilt had developed after the injuries. 


*I don’t know when I started to think that it was my fault.*
(Parent 21)


*I afflicted my completely smart, healthy, bright little child with an injury for life, and I can’t redo it. A small healthy child, whom I voluntarily gave a medication which harmed him. That is the worst. For me, it would have been a million times easier to accept, if he had had the same injuries from measles, or mumps, or rubella or something else. Because this is something that I have done.*
(Parent 21)

The feelings of guilt had not been resolved. Moreover, the injuries had affected the entire families, including the healthy children.


*The younger brother he has felt ignored,—and he still does.*
(Parent 24)

These parents were not opposed to vaccinations. However, they emphasized the need for information about the risk of injury and for more responsibility on the part of pharmaceutical companies for developing safe vaccinations and the community for helping families with injured children. They both opposed moving MMR1 to 6 months of age, primarily due to concern that side effects might be less visible in younger children. They explained how the sudden disappearance of abilities their child had already developed was an obvious sign of injury.


*When the fever disappeared, it was that HE also disappeared. Then, the light in his eyes was simply turned off. He disappeared completely, and just looked confused.*
(Parent 22)


*I can be a little worried about whether side effects are discovered in children who do not have those developments (…) Well, our girl could walk, and she could talk, she could sing. That is, when she got the injuries. If you do this [vaccinate at 6 months] … will you then discover those side effects?*
(Parent 24)

### 3.4. HCPs Support MMR Vaccination and Are Careful Not to Lose Contact with Reluctant Parents

All HCPs supported both the MMR vaccine and advancing MMR1 administration to 6 months. They challenged parents who were indecisive, reluctant, or opposed to MMR vaccination. Still, they were cautious about not losing contact with parents who needed their support in other areas. In particular, HVNs emphasized that they were careful to maintain the relationship with the family in the way they supported parents in accepting the vaccine. Thus, they seemed to avoid the subject with parents who were fully determined not to have their child vaccinated and might show defensive vaccine rejection. This was explained by two HVN: 


*When they say almost aggressively: ‘We have taken a stand on it. We will not discuss it’. Then I don’t give it many attempts.*
(HCP 2)


*When you sort of avoid the discussion, that is with a total vaccination opponent, then it is to keep the good relationship.*
(HCP 1)

An exception to this finding was a GP who described how he was openly challenging parents, raising their awareness to the fact that choosing NOT to vaccinate their child was also a choice. 


*We [the employees in his general practice] will not choose on behalf of people, but we like to express our clear attitude and at the same time respect their attitude. The thing about simply not wanting to vaccinate at all—we just question that.*
(HCP 10)


*Parents in his practice rarely rejected the MMR vaccine; he could only recall one family among his 9000 patients that had ultimately rejected the vaccine. When asked what people might fear, he mentioned that they often worried whether the vaccine might inhibit the natural immune system. He hypothesized that rejecting the vaccine might be an attempt to create a “a type of false security, about something that you actually can’t be so sure about”.*
(HCP-10)

Moreover, he underscored that the severity of the diseases prevented by the MMR vaccine has been forgotten, pointing at the need for information as a contributing factor in decision making.


*But uh, they cannot remember any epidemics or seriously ill or deaths. So, you know more about vaccination injuries than you know about diseases.*
(HCP-10)

Recalling the severity of the diseases made one HVN exclaim:


*I tell [the parents] what it’s like to have mumps, rubella and measles. I can remember them all.*
(HCP 4)

Among HVNs, the level of information provided depended on the questions and desires of the parents. However, all HCPs reported that most parents only requested information about immediate side effects.


*Then you say sort of… ‘there can be a local reaction, and they might get a little fever in the evening’.*
(HCP 8)


*I don´t hand out the pamphlet [about vaccination] to everybody. Only the ones who have doubt concerning dates and so on. The…need to have it in writing, right?*
(HCP 3)

GPs in particular predicted a positive impact of moving MMR1 to 6 months instead of 15 months. One GP expressed it as follows:


*It is a good window [at six months] to do it. There is even greater compliance in the very first part when the mother has just given birth, she has been here [at the GP] so frequently and the family is still preoccupied with this new [child] in very new ways…*
(HCP 10)

This GP expected that parents might be at greater risk of forgetting the vaccine at 15 months.


*They become more careless about showing up and stuff like that. …I think it could promote compliance that the child is so small [at six months].*
(HCP 10)

## 4. Discussion

The primary study findings are the four parental attitudes about current MMR vaccines and advancing MMR1 administration that are consistent with their trust in the healthcare system, from unquestioning trust to outright rejection. Low mutual tolerance between vaccine supporters and opponents was reflected by opponents joining a vaccine-skeptical group. Parents of children seriously injured by a previous MMR vaccine felt let down and described lifelong unresolved guilt. Finally, HCPs’ responses to the four parental attitudes revealed their caution about not losing contact as reason to avoid arguing with defensive parents. Current MMR vaccine supporters emphasized that it is a well-known vaccine and advancing MMR1 administration will be timely and convenient, whereas opponents argued it disturbs a child’s immature immune system and emphasized the challenge of detecting side effects earlier in life. It is notable that HCPs predicted unchanged or higher vaccine coverage when administering MMR1 at 6 months of age.

The pending results of the MMR trial will indicate whether administering MMR1 at six months of age is recommended. Whether or not that is the case, our findings provide researchers and clinicians with important new knowledge about the various needs at play among people with different viewpoints on MMR vaccination and advancing MMR1 administration. Our findings indicate that parents who are unquestioningly trustful will do well with a possible advance in MMR1 administration if provided with information about the rationale for and potential consequences of the change. 

In contrast, parents who accept MMR vaccines after careful consideration appear to need expanded knowledge about complications related to the diseases the vaccine prevents. This need, expressed both by parents and HCPs, will likely increase over years as knowledge about the severity of measles, mumps, and rubella continues to fade, leaving an unbalanced focus in the literature on the side effects of vaccination. This fails to provide an adequate foundation for parents weighing the reasons for and against getting their child vaccinated. Providing access to this information, as well as side effects of the MMR vaccine, presents an important step for the healthcare system toward empowering parents to facilitate self-determined choices about childhood vaccinations. This indicates the need for easily accessible information, such as a short film, that can enlighten parents about the MMR vaccine and the fact that, before the introduction of the MMR vaccine, 10–15 children a year in Denmark suffered from severe encephalitis due to measles and, in 1960–1986, an average of 3.3 children died each year due to the virus [6,26]. Short films informing parents about children’s health issues is a well-known communication method in Denmark [27].

This strategy may also support vaccine-hesitant parents to make an informed choice, rather than the “non-choice” a participant described. Suggested initiatives to help parents who are vaccine hesitant or reluctant include clearer communication strategies, establishment of a therapeutic alliance, health literacy, and empowerment [28,29]. Our finding that parents who lingered in indecision had no one to talk with about their doubts and felt they did not belong anywhere is likely related to the identified mutual intolerance between supporters and opponents. For this group of parents, HCPs play a major role in establishing a therapeutic relationship encompassing respect for people’s choices, affirming their values, and protecting their rights [28]. 

The need for therapeutic relationships is also suggested by our finding that several parents who expressed doubt or resistance towards vaccinations described HCPs responding to their concerns with mistrust and directives about what to do. In contrast, respectful curiosity and a sincere desire to support parents’ decision making may prevent the development of the defensive attitude that HCPs found most difficult to address. A perceived lack of interest among HCPs in understanding the people’s arguments against and concerns about vaccinations may cause further resistance to and mistrust of the healthcare system in general, resulting in lower vaccine coverage [30,31]. 

Another option is introducing an informed refusal tool [30,32]. This may also be helpful in cases where vaccine refusal is related to conspiracy theories proposed by people who refuse vaccination. 

This analysis also includes parents of children with perceived serious vaccine-related injuries to represent the ultimate, albeit rare, negative consequence of childhood vaccinations and reveal its effect on their entire lives. The lifelong feelings of guilt which can be harmful both for parents and the whole family seemed to be neither understood nor remedied by healthcare services. Despite the challenges they continued to face, these parents did not oppose vaccinations, calling instead for more information from the healthcare system about potential serious side effects and help for affected children and families. 

This study took place during the COVID-19 pandemic, which generated a new and somewhat more open discussion about vaccinations and their acceptance and rejection. Researchers in COVID-19 vaccines have highlighted the childhood vaccination program as a successful program to learn from to increase vaccine uptake [33]. Conversely, we argue that the MMR vaccine program has something to learn from the COVID-19 vaccination program about procedures for systematically reporting side effects [34,35]. 

### Strengths and Limitations

Our study has several strengths. Triangulating perspectives from parents and healthcare professionals yielded a broad view of MMR vaccination and the timing of MMR1 administration. Recruiting parents through different pathways resulted in a diverse group of participants drawn from the general population, increasing the likelihood that they are representative of the Danish population as a whole. All interviews and primary analysis were conducted by the same two researchers, and final analyses and thematic coding were validated with a third author. Study limitations are primarily due to the difficulty of recruiting parents with doubt about or resistance towards MMR vaccination despite various attempts. This reflects a generally high acceptance of vaccinations in Denmark, but it may also limit the generalizability of our findings to all Danish vaccine-averse parents. Finally, due to COVID-19, most interviews were conducted online, but we view in-person communication as optimal.

## 5. Conclusions

This trial-embedded qualitative study provided new insight into parents’ and healthcare professionals’ attitudes about MMR vaccination in general and about advancing MMR1 administration. We identified four attitudes that were consistent with parents’ trust in the healthcare system and mutual intolerance between vaccine supporters and opponents. Parents of two children who they believed had been seriously injured in connection with MMR vaccination approximately 15 years previously revealed unresolved guilt and the need for more psychological support. Current MMR vaccine supporters viewed advanced MMR1 administration as positive, expecting it to be more convenient, whereas opponents feared it might disturb a younger child’s less mature immune system and make it more difficult to detect side effects. It is notable that HCPs predicted unchanged or higher vaccination coverage if MMR1 administration was advanced to 6 months of age.

To specify how widespread the different attitudes are, regarding developing implementation strategies for decision makers, further quantitative studies are mandatory. This study presents important background information for further research in the field.

## Figures and Tables

**Table 1 vaccines-11-00067-t001:** Characteristics of participating parents and their children.

Number	Gender	Age (Years)	Age of Child	Number of Children in Family/ Gender of the Child	MMR Trial Participant	Interview Format
1	F	37	5 mo.	3/M	Yes	Face-to-face
2	F	34	6 mo.	2/F	Yes	Face-to-face
3	M, F	37, 33	5 mo.	1/F	Yes	Face-to-face
4	F	43	5 mo.	3/M	Yes	Face-to-face
5	F	30	6 mo.	2/F	Yes	Face-to-face
6	F	26	5 mo.	1/F	Yes	Face-to-face
7	F	32	7 mo.	1/F	No	Online
8	M	34	15 mo.	1/M	No	Face-to-face
9	F	40	22 mo.	3/M	No	Telephone
10	M, F	30, 32	6 mo.	2/M	Yes	Face-to-face
11	F	36	12 mo.	2/M	No	Telephone
12	F	29	16 mo.	1/M	No	Face-to-face
13	F	34	11 mo.	2/F	Yes	Telephone
14	F	32	17 mo.	2/M	No	Online
15	F	24	20 mo.	1/M	No	Online
16	M	31	8 mo.	1/F	No	Face-to-face
17	F	33	14 mo.	1/M	No	Online
18	F	35	2 mo.	4/M	No	Face-to-face
19	F	28	11 mo.	1/F	No	Telephone
20	F	41	12 mo.	3/M	No	Telephone
21	F	67	28,* 33, 38 years	2 */M	No	Telephone
22	F	59	5, 7, 19, 27 years	-	No	Telephone
23	M	49	4, 9 years	-	No	Face-to-face
24	M	49	13, 18 *, 21 years	2 */F	No	Telephone

* Previous serious vaccine-related injury.

**Table 2 vaccines-11-00067-t002:** Characteristics of participating healthcare providers.

Number	Gender	Profession	Interview Format
1	F	HVN	Individual
2	F	HVN	Individual
3	F	HVN	Individual
4	F	HVN	Individual
5	F	HVN	Dyad
6	F	HVN	Dyad
7	F	GP	Focus group
8	F	GP	Focus group
9	F	GP	Focus group
10	M	GP	Individual
11	F	GP nurse	Individual

**Table 3 vaccines-11-00067-t003:** Parental attitudes about MMR and advanced MMR1 administration and HCPs’ response.

Parental Attitudes about MMR Vaccines	Unquestioning Trust	Acceptance after Careful Consideration	Challenging Indecisiveness	Rejecting in A Defensive Way
Parent trust in the healthcare system	Unlimited	Nuanced and informed	Limited	Absent, with tendencies toward conspiracy theories
Sources of information	Do not seek information Family history GP Leave it to experts	National Board of Health Serum Institute Danish Medicines Agency	Questioning information from: National Board of Health; Serum Institute; Danish Medicines Agency.	Mistrust towards: National Board of Health and Serum Institute. Uses: Alternative sources International literature
Challenges	None	None	Feeling bewildered, lonely, and not belonging. Difficulty making decisions on behalf of one’s child	Feeling overruled and confronted
Parental attitudes about advancing MMR1 administration	Positive	Positive due to known vaccine Expect sufficient information	Negative due to child’s immature immune system	Highly negative due to child’s immature immune system and difficulty of discovering side effects in young children
HCP response to parent attitude	Supportive	Supportive, emphasizing that forgotten disease severity may impede the ability to weigh for and against	Challenge reluctance HVNs cautious because parents still need their service GP stresses that avoiding vaccines is also a choice	HVNs avoid arguing with parents who have made their decision in a defensive way

## Data Availability

Not applicable.

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
