# Peer review of "Parents’ and Health Professionals’ Attitudes to Advancing Primary MMR Vaccine Administration from Fifteen to Six Months of Age—A Qualitative Thematic Analysis Embedded in a Randomized Trial"

_vaccines, 2022, doi:10.3390/vaccines11010067_

Round 1

Reviewer 1 Report

A very interesting, informative and well written manuscript that has clinical merit.  However, there are some editing issues that the authors should address.  The following are suggestions/comments regarding those issues.  Lines 18 & 19, "...prompted a Danish trial to test the feasibility ...".  Line 21, "...professionals" (N=11) attitudes about measles, mumps, and rubella vaccine (MMR) in general ...".  Lines 23 & 24, "...parents considering that their child might sustain a serious vaccine-injury."  Line 30, "...feared it would disturb the children's immature immune ...".  Line 46, "....NCT03780179]) enrolled 6500 infants and examined ...".  Line 49, "...effects of the vaccine, which would be published in 2022-2024."  Line 54, "crucial in stopping the spread of measles."  Line 59, "...for a four-year plan involving new immunization ...".  Line 138, "...in a focus group, a dyad and ...".  Line 139, "...(Table 2) at the participants' request."  Lines 142 & 143, "...medical examination, a vaccination and were fatigued."  Lines 227 & 228, "...particularly from the NBH home page and ...".  Line 304, "...expressed disbelief in the information from the ...".  Lines 403 & 404, "...administration at 6 months."  Line 409, "...not to have their child vaccinated and ...".  Line 502, "...HCPs in understanding the people's arguments ...".  

Author Response

Thank you.

In the revised manuscript all these comments have been meet except for lines 403 and 404, where we preferred to keep "advancing MMR1 administration to 6 months"

Reviewer 2 Report

This paper presents a qualitative study about social dynamics in a population of possible users and family members involved in the decision process concerning vaccination program.

The topic is interesting and the amount of information is useful. On the other hand, is seems to me that the paper is not complete as it does not yet lead to select specific strategies. Indeed, decision makers need a filter toward the interpretation of the data. This filter ca be obtained by mathematical models focused on social dynamics.

In my opinion, this paper even if does not develop this matter, which would amount to write a new paper, should mention the problem. The open access paper in the following provide an address to the literature on social dynamics. The authors are NOT required to cite this paper, but grasp in the bibliography to understand is some indications are useful.

https://www.worldscientific.com/doi/pdf/10.1142/S0218202521500408

Author Response

Thank you so much for your review report. 

We haven’t found it relevant to include a mathematical model concerning social dynamics due to the limited number of participants in our qualitative study. But as advised, we have commented on this issue in our conclusion, by adding the following sentence:

“To specify how widespread the different attitudes are, regarding developing implementation strategies for decision makers, further quantitative studies are mandatory. This study presents important background information for further research in the field”

Reviewer 3 Report

This study examined an important health issue. I have some suggestions for the manuscript.

Title

The title can be shortened into “Parents’ and health professionals’ attitudes to advancing primary MMR vaccine administration from fifteen to six months 3 of age.”

Abstract

“Health professionals were cautious about losing contact with vaccination-reluctant parents.” The meaning of this sentence is unclear.

Introduction

l   “…due to declining levels and duration of passively ac- 41 quired maternal antibodies…” Was there scientific evidences to support that?

l   The sentence “Beginning in April 2019, a double-blind,…will be published in 2022-2024” in the first paragraph should be integrated into the second paragraph.

l   “awareness of vaccine acceptance or rejection has increased recently due to the COVID-19 pandemic” I am not sure how the COVID-19 pandemic increased the awareness of vaccine acceptance or rejection.

Methods

l   “…the long-term perspectives of two parents whose children were seriously injured in connection with MMR vaccination 27 and 17 years before.” It will be clear to describe what kinds of long-term injuries connected with MMR vaccination and how the connections were determined.

l   “…but the COVID-19 pandemic necessitated conducting many by videoconference or telephone.” The meaning of this sentence was unclear.

Results

“Low tolerance between vaccine supporters and opponents” Tolerance about what? It should be illustrated.

Author Response

Thank you for your review report. 

We have with interest read your comments and have replied accordingly. Please read our replies and changes in the attached document.
